# The Analysis of Facio-Dental Proportions to Determine the Width of Maxillary Anterior Teeth: A Clinical Study

**DOI:** 10.3390/jcm11247340

**Published:** 2022-12-10

**Authors:** Naseer Ahmed, Mohamad Syahrizal Halim, Zuryati Ab-Ghani, Maria Shakoor Abbasi, Ayesha Aslam, Jawad Safdar, Gotam Das, Abdul Razzaq Ahmed, Nafij Bin Jamayet

**Affiliations:** 1Department of Prosthodontics, Altamash Institute of Dental Medicine, Karachi 75500, Pakistan; 2Prosthodontics Unit, School of Dental Sciences, Health Campus, Universiti Sains Malaysia, Kota Bharu 16150, Malaysia; 3Conservative Dentistry Unit, School of Dental Sciences, Health Campus, Universiti Sains Malaysia, Kota Bharu 16150, Malaysia; 4Department of Prosthodontics, Army Medical College, Armed Forces Institute of Dentistry, National University of Medical Sciences, Islamabad 44000, Pakistan; 5Department of Oral and Maxillofacial Surgery, Dow Dental College, Dow University of Health Sciences, Karachi 74200, Pakistan; 6Department of Prosthodontics, College of Dentistry, King Khalid University, Abha 61421, Saudi Arabia; 7Division of Restorative Dentistry, Prosthodontics, School of Dentistry, International Medical University, Kuala Lumpur 57000, Malaysia

**Keywords:** golden proportion, RED proportion, horizontal facial proportion, central incisor width, intercanine distance, dental photography, 3D dental analysis, dental cast, dental arch symmetry

## Abstract

The present study aimed to analyze mid horizontal facial third proportions, those being the interpupillary, inner intercanthal, and bizygomatic distance modified with golden proportion, The Preston proportion, golden percentage and 70% recurring esthetic dental proportion were used for determining maxillary anterior teeth width. A total of 230 participants took part in this study. The front dental and facial photographs along dental stone cast which were converted to three-dimensional (3D) models were used for evaluation. The mid horizontal facial third proportions showed no significant relationship with maxillary anterior teeth width without modification with dental proportions. Whereas, with modification, no statistically significant difference was found between inner-intercanthal distance by golden percentage and width of central incisors. The bizygomatic distance was greater than intercanine distance. While the interpupillary distance by golden proportion was found to be consistent with intercanine distance in female participants. The modified anterior teeth width was significantly different from measured values, when determined by using the three mid facial proportions with Preston and 70% recurring esthetic dental (RED) proportion. Furthermore, the measured width of maxillary anterior teeth showed no difference when plaster dental casts widths were compared with 3D models. The interpupillary, inner-intercanthal, and bizygomatic distance should not be directly used to determine maxillary anterior teeth width. While maxillary anterior teeth width can be determined by modifying the inner inter-canthal distance with golden percentage and interpupillary distance with golden proportion. Moreover, the midfacial third proportions modified with Preston and 70% recurrent esthetic dental proportion were found to be unreliable for the determination of maxillary anterior teeth widths.

## 1. Introduction

In an attractive smile, the maxillary anterior teeth play a pivotal role. The destruction and loss of anterior teeth can cause psychological trauma and deterioration of facial esthetics. Hence, in order to achieve facio-dental harmony, appearance and functional restoration of smile should be provided [1]. Though, smile preferences are a regional phenomenon. Dimensional differences exist between different populations based on sex, race, and ethnicity. For example, in Asia the lateral incisors are often narrower than in Africa, and here is often a diastema between the central incisors which affects the proportion. This affects smile preferences and proportion of teeth in different areas of the world. Hence, research should be developed and considered specific to a particular group of people [2].

The tooth size selection is important when natural teeth are to be replaced. Artificial teeth should conform to the overall facial appearance and allied oral structures [1,2]. Generally, the restorative dentist follows the width and height ratio of anterior teeth in smile design [3]. The determination of teeth width is a cumbersome procedure due to the complexity of calculation methods, compared to the tooth height determination, because smile arc, smile line, vertical facial heights, and lip length are the reliable reference points for maxillary anterior teeth (MAT) length determination [4].

Over a period of time, various horizontal facial proportions were used to determine the mesiodistal width of MAT. Within the facial anatomical reference points, one can approximately replace the MAT, but this has been complicated by the variation in esthetic preference and facial features [5]. Nevertheless, according to dentofacial esthetic principles, the interpupillary distance (IPD), inter-alar distance (IAD), and bizygomatic distance (BZD) are generally used to determine the central incisors width (CIW) and inter-canine distance (ITCD) [6].

An approximate match of IPD, BZD, and ICD to ITCD and CIW has been proposed in scientific literature. The ITCD is said to be equal to IPD and BZD, while inner intercanthal (ICD) to central incisors width, which makes these facial proportions a reliable parameter to assess the mesiodistal width of MAT [6,7]. However, evidence contradicting the reliability of these facial proportions as an aid to determine the MAT dimensions also exists in the literature [8,9].

The MAT combinations, CIW, and ITCD need to be in harmony with facial form and features to appear attractive. The recurring esthetic dental (RED) proportion [10,11], golden percentage or mean (GM) [12], golden proportion (GP) [11], and Preston proportion (PRP) [13] are endorsed as parameters to determine the mesiodistal width of MAT. In the literature, a proportional relationship of GP, GM, and RED with facial dimensions has been reported [14,15]. The GP was the first to be introduced in dental esthetics by Levin in 1978 [11]. The GP theory explained a mathematical and natural relationship in beauty where the teeth are in a proportion of 62% to the facial dimensions and to each other in the dental arch when viewed from the front. Additionally, the GM, which is another dental esthetic proportion theory was proposed by Snow in 1999 [12]. The GM theory states that the central incisor width is 25% of the combined six anterior teeth width while the lateral is 15% and the canine is 10%. The PRP was proposed by Preston in 1993 to attain attractive MAT composition. It states that the teeth vary in size, and therefore the dental ratio cannot be constant to 62% as proposed by Levin [11]. Instead, Preston proposes that the lateral incisor should be 66% of the central incisor width and the canine 84% of lateral incisor width [13]. The PRP was evaluated in the width of MAT through the direct and perceived measurement of teeth utilizing dental cast and 2D photographs. It was rarely found in the width of MAT [15,16]. However, as far as the determination of CIW and ITCD through the modification of mid-horizontal facial proportion (MHFP) is concerned, it is yet to be studied.

Recently, an improved version of dental proportion was proposed by Ward in 2000, the recurring esthetic dental proportion, which suggested that the ratio of anterior teeth remains constant when one moves distally in the dental arch. The lateral incisor’s width should ideally be 70% of central incisor whereas canine’s width should be 70% of lateral incisor tooth. The ratios should be constant, while the values can vary according to the width and height ratio of anterior teeth [17]. Therefore, a tall tooth will have a different constant proportion while a small or medium tooth may have different width ratios. The use of this theory to determine CIW and ITCD from mid horizontal facial proportion (MHFP) modification is lacking.

A spectrum of dental proportion theories had been suggested to evaluate the width of natural MAT. Prediction of CIW and inter-canine distance utilizing these dental proportions are yet to be explored and supporting evidence is required. Therefore, this study aimed to compare the IPD, ICD, and BZD with GP, PRP, GM, and 70% RED proportion in order to determine the inter-canine distance and the central incisors’ width. The hypothesis stated that IPD, ICD, and BZD could serve as reliable means for assessing MAT width by modification with GP, PRP, GM, and 70% RED proportion.

## 2. Materials and Methods

### 2.1. Study Setting and Estimation of Sample Size

This analytical study was conducted at Altamash Institute of Dental Medicine (AIDM), Pakistan comprising of 230 participants ranging from 18 to 30 years of age. Non-probability convenience sampling method was adopted to recruit participants according to a set subject criterion. The sample size was calculated with, “Creative research systems survey software (Creative research systems, version 9, Petaluma, CA, United States).” Considering the mean ICD of 30.48 ± 2.01 [18], it kept alpha error at 5% and a confidence level of 95%. The estimated sample size of 230 was calculated (considering the 10,000,000 population). In this study participants with intact, natural MAT were included.

### 2.2. Participant Enrollment and Ethical Consideration

After obtaining ethical approval from the Institute’s Ethical Review Committee (AIDM/EC/06/2019/06) and written informed consent from the participants, the demographics were recorded in a data collection form, including nationality, height, and weight. The weight was recorded with a digital scale (Seca digital flat weighing machine, China) in kilograms (kg) while a stadiometer (Seca 224 conventional meter, China) was used to record height in centimeters (cm). The participants then underwent extraoral examination in order to rule out any facial deformity, asymmetry, temporomandibular disorder, and restricted mouth opening. Intraorally, they were examined for the presence of any carious lesions or restorations in anterior teeth, malalignment and gingivitis. Out of the initially recruited 250 subjects, 20 were excluded based on the presence of maligned teeth, asymmetries of face, prior restorative treatment, i.e., composite restorations, history of orthodontic treatment, crown and bridge work, distorted/blurred photos, impression errors, and fractured or faulty dental casts.

### 2.3. Capturing Dental and Full Face Frontal Photographs

A digital single lens reflecting camera (“Canon EOS; CMOS; 18 MP,1920 × 1080 p/30 fps”) set at 12 o’clock position on a tripod at a distance of 1.5 m was used to capture high-resolution photographs of teeth and full-face with 1:2 macro setting (intraoral) and 1:10 (Extraoral), respectively. The ISO was set from 100 to 200 depending on the lighting conditions. The aperture was set to f-20 for intraoral and f-8 for extraoral images. The shutter speed was set to 1/125 sec in both views. A ring fluorescent light source system (LED-FD,480II; Medike Photo and Video Co. Ltd., Yidoblo, Guangdong, China) comprising of a light unit mounted adjacent to the camera lens was used. A color indicator dot was placed on the forehead of the subject to judge the photographic error or image distortion. Full-face photographs and anterior teeth images were taken from the front, with the subject seated comfortably upright and head held straight and facing forward in a natural position. The camera lens was fixed at the subjects’ eye level for full-face images, while for retracted smile images, the lens was adjusted at the incisors’ level. For all intraoral photographs, lips were retracted to clearly exhibit the MAT. The protocol was similar to the study carried out by Bidra et al. [19]. The 2D photographic width of anterior teeth was measured from the labial side. The perceived width of anterior teeth was measured in a straight line which a pre-requisite for determining dental proportion. Moreover, the intercanine distance was also calculated in a straight line between the most apparent mesial and distal points of teeth. As shown in Figure 1.

### 2.4. Registration of Dental Impression and Fabrication of Dental Stone Cast

In order to record the impressions, perforated stainless-steel maxillary impression trays extending up to the hamular notches and fovea palatinae were selected. It was ensured that the borders of the tray covered the functional sulcus depth within physiological limits and that a uniform space of 3–4 mm existed for the impression material between the tissues and tray flanges.

For all subjects, the maxillary arch impressions were made with irreversible hydrocolloid impression material “(Fast setting alginate Hydrogum, Zhermack SpA, Badia Polesine, Italy)”. After the disinfection, each impression was marked with an identification number. Dental stone casts were fabricated by pouring the impressions with Type IV dental stone “(ISO Type 3, Elite Rock Zhermack SpA, Badia Polesine, Italy)”. In order to eliminate errors such as dimensional variations and desiccation, stone casts were removed after 30 min of pouring and were serial-coded using a permanent marker. Standard base formers were used to fabricate soft plaster bases for all casts. Three-dimensional models were obtained by scanning the dental stone casts with a desktop 3D Dental Laboratory Scanner “(UP360+, 300 × 300 × 400 mm, 3D scanner, Shenzhen, China)”, a high precision scanner that employs 2.0-megapixel cameras. Scanned 3D full-arch images were displayed on a compatible dental design software (UPCAD, UP3D, Shenzhen, China), then transferred and stored on a personal computer.

### 2.5. Dental Stone Cast and 3D Model Analysis

The mesiodistal dimension, i.e., the width of MAT on the 3D model was documented in millimeters using the measurement tool on Photoshop software (Adobe, version 21.0.2, San Jose, CA, USA). Furthermore, the widths of teeth on the dental stone casts were measured with a digital Vernier caliper to the nearest 0.02 mm. The zero error of the caliper was adjusted during calibration. The minus error (−) value for if anything was added to the readings was noted, while a positive error (+) was subtracted from the vernier caliper reading. The widths of incisors and canines were measured from the facial aspect using the outer edges of the Vernier caliper positioned between the contact points of each tooth (Figure 2).

### 2.6. Horizontal Facial Third Proportion Calculation

The widths of the middle third of the face between anatomical reference point IPD, ICD, and BZD (Figure 3) were measured on the frontal facial photographic images via Adobe Photoshop software (Adobe, version 21.0.2, San Jose, CA, USA). The BZD was measured as “the distance between lateral borders of right and left zygoma”. The ICD was obtained as “the distance between medial angles of palpebral fissures”. Meanwhile, IPD was acquired as “the distance between the right and left pupils of the eyes, looking straight ahead”.

### 2.7. Data Validity, Reliability, and Photographic Error Assessment

The data have been collected by a single operator (N.A.). In order to calibrate the examiner, the facial and dental measurements were initially recorded by a senior operator (J.S.). A correlation analysis was then performed using the measurements obtained by the two operators. A strong correlation value of 0.739 was revealed. Additionally, 20% of the data including photographs and dental models were re-evaluated two weeks later by the same operator, and data were analyzed with a correlation statistic to assess intra-operator reliability.

For data validation, 20% of the dental stone cast and photographic measurements that had been made manually using a vernier caliper were compared to 3D model measurements recorded via Adobe Photoshop software. Association between the two datasets was evaluated using the “intraclass correlation coefficient test (ICC)”, and a strong correlation value of (0.816) was found.

The photographic error was minimized by obtaining a conversion factor. This conversion factor was achieved by dividing the actual MAT width of dental stone casts by the perceived width calculated from photographs [10]. When the perceived maxillary teeth widths were multiplied by the conversion factor, it helped in eliminating the magnification error and producing the true teeth width. This modified maxillary mesiodistal teeth dimension obtained after photographic error estimation assessment was named clean width in this study.

### 2.8. Statistical Analysis

Data analysis was performed via Statistical Package for the Social Sciences Software (IBM, SPSS Statistics, version 25, Chicago, IL, USA). Shapiro–Wilk test along with normality plots was used to evaluate the normal distribution of data. A descriptive analysis of categorical (gender) and continuous (age, height, weight, teeth widths, facial proportions) variables was performed to calculate the frequency, percentage, mean and standard deviation. Moreover, regression analysis, independent *t*-test, and paired *t*-test were used to compare the mean values of dependent (MAT, midfacial horizontal facial proportion) and independent (age, gender, height, and weight) variables. A *p*-value of ≤ 0.05 was considered statistically significant.

### 2.9. Predicting ITCD and CIW from MHFP Modification by Dental Proportions

In order to predict inter-canine distance, the IPD, BZD, and ICD measured values were multiplied with set values of dental proportion theories (62% golden proportion, 70% RED, 66%, and 84% Preston proportion, and 25%, 15%, and 10% golden percentage).

Moreover, in order to predict the central incisors’ width, the mid-facial proportions values were multiplied with dental proportion ratios (70% RED proportion, 0.5% GM, 1.618% GP, and 1.32% PRP. The metrics adopted to determine inter-canine distance and central incisors width via modification of mid-facial third proportions by dental proportion is mentioned in Table 1.

IPD: Interpupillary distance, ICD: Inner-intercanthal distance, BZD: Bizygomatic distance.

## 3. Results

The current study included 230 participants with a dropout rate of 0.08% and a mean age of 24.210 ± 3.541. Out of the total, 112 (48.7%) were males and 118 (51.3%) were females. The mean height of participants was 168 ± 14.84 cm while the mean weight was 65.93 ± 13.1 kg.

The mean widths of MAT obtained through 2D dental images, 3D models, and dental stone casts are shown in Table 2. No significant difference was observed between the ITCD achieved from plaster and 3D dental models (*p* = 0.073).

The clean widths of MAT are shown in Table 3. The difference between the mean values obtained via 2D photographic and clean widths of MAT was statistically significant (*p* < 0.05), as shown in Table 3.

The analysis of gender disparity in mean MAT width obtained from 3D dental models showed a significant difference (*p* = 0.022) between the mean values of right lateral incisor in both sexes. The mean values of the right central incisor (*p* = 0.138) and canine (*p* = 0.502) did not show a significant difference, respectively. Similarly, no significant difference was found in the left central incisor (*p* = 0.053), left canine (*p* = 0.361), and left lateral incisor (*p* = 0.700), respectively. Additionally, no significant difference (*p* = 0.531) was observed in the intercanine distance of both sexes. Although, a significant difference was found in RLI teeth in both sexes, which was indicated by a small *t*-value of (−2.305) and a mean difference of (−0.105) between the widths of this tooth, Table 4.

Comparing the clean mean MAT width in both sexes showed a significant difference for mean RLI (*p* = 0.043) and RCa values (*p* = 0.004) while no statistical difference (*p* = 0.216) was found between the values of RCI. Similarly, a significant difference in mean values of LCI (*p* = 0.053) and of RCa was also different (*p* < 0.001). However, no significant difference was found in mean values of LLI. Additionally, the difference in intercanine distance between males and females was statistically significant (*p* < 0.001). In addition, a significant difference was seen in RCI and LI teeth also in LCa and RCa in both sexes, it was indicated by a small *t*-values and mean differences as shown in Table 5.

The mid-horizontal facial proportions (MHFP) analysis performed over 2D photographs is shown in (Table 6).

The gender disparity in mean mid-horizontal facial proportions values is shown in Table 7.

The MHFP was modified to determine the central incisors’ width and inter-canine distance using dental proportions in both sexes as shown in Table 8. When evaluation of midfacial dimensions was performed by 70% RED proportion, GM, and PRP, the mean values of IPD were significantly (*p* ≤ 0.001) larger than the intercanine width. While the IPD by golden proportion value was significantly smaller than ITCD. The mean values of ICD were significantly (*p* < 0.001) smaller than the intercanine distance. However, the mean values of BZD were significantly (*p* < 0.001) larger than the intercanine distance Table 8.

A separate analysis was carried out for the male and female groups to assess the MHFP with dental proportions to determine CIW and ITCD as shown in Table 9 and Table 10, respectively.

The mean IPD and BZD with GM modification values were greater than central incisor width. However, the mean ICD and CIW had an exact match. The IPD showed significantly larger values with all dental proportions compared to inter-canine distance (*p* < 0.001) except for the golden proportion which had a smaller value. Similarly, BZD yielded significantly (*p* < 0.001) greater values than inter-canine distance, whereas the ICD mean values were significantly (*p* < 0.001) smaller than inter-canine distance, Table 9.

The analysis of MHFP by 70% RED proportion, PRP, and GP in females yielded significantly larger mean IPD, ICD, and BZD values than the central incisors width (*p* < 0.05). In addition, mean IPD and BZD values with golden percentage were significantly (*p* < 0.001) greater than the CIW. An exact match of ICD and CIW values was seen in females (*p* ˃ 0.05). MHFP analysis with 70% RED proportion, PRP and GM produced significantly greater mean IPD and BZD values than inter-canine distance (*p* ≤ 0.001) whereas the mean ICD value was significantly smaller (*p* ≤ 0.001). The mean IPD with GP value was similar to ITCD (*p* ˃ 0.05) in females, Table 10.

The regression analysis of MHFP and independent variables (height, weight, age, and sex) are shown in Table 11. The analysis revealed a weak correlation between MHFP and the independent variables. The IPD analysis revealed a R-Squared (R^2^) = 0.041 and an adjusted R-Squared (AR^2^) = 0.0241. However, under the influence of external variables only gender presented a significant difference (*p* = 0.014). While the (R^2^) for ICD was 0.014 and AR^2^ = −0.004, the correlation of independent variables with ICD was insignificant (*p* ˃ 0.05). For BZD, R-Squared (R^2^) = 0.023, and the adjusted R-Squared (AR^2^) were 0.006. However, under the influence of external variables only weight presented a significant difference with BZD (*p* = 0.035).

The IPD to gender beta (B) value was statistically significant (B= −0.182, *p* = 0.014) which indicates that 18.2% variation in IPD can be attributed to gender whereas on average the effect of gender on IPD was B_o_=4.834 in this study. Moreover, weight to BZD beta value was also statistically significant (B = −0.154, *p* = 0.035) which showed that with an increase in weight the BZD value was increased by −0.154 whereas the average effect of weight on BZD was B_0_ = −0.114 in this study.

## 4. Discussion

The present study is novel in being the first orofacial anthropometric research comparing MHFP with theories of dental proportions to predict the width of MAT. Additionally, this study proposes valid metrics to determine the width of anterior teeth. Furthermore, the ICD by GM and IPD by PRP modification can reliably predict CIW and ITCD in both sexes. While on the other hand, inter-pupillary, inner canthal, and bizygomatic distances without modification by DP do not correlate to MAT width in this study. This finding was similar to Bidra et al. [19] and Godinho, J. [20]. On the contrary, the exact match of ICD with CIW in both males and females was an unexpected finding. The present study found equal values in both sexes when MAT width was evaluated with ICD-GM and IPD-PRP combinations in females. This, however, is in contrast to the general belief of “gender disparity” influences the estimation of anterior tooth dimensions [21,22]. This difference in findings of the present study could be attributed to bias in sample size calculation and overestimation of the results of reported studies.

In the present study, mean width of MHFP by DP was significantly greater than the CIW in both sexes, except for ICD by GM values. The bizygomatic distances with all 4 dental proportions, whereas inner canthal and interpupillary distance with 70%RED, GP, and PRP values were different CIW in the participants. Therefore, these dental proportions were considered unreliable in determining the combined CIW.

Moreover, regarding the intercanine distance, inter-pupillary and bizygomatic distance values modified by dental proportions were found greater than ITCD in both sexes. Only the female cohort of IPD-by-GP combination showed no statistically significant difference. Overall, the modified ICD values were significantly smaller than ITCD. Keeping this in view, it can be implied that MHFP showed varying values in the participants of this study, thereby making ICD reliability questionable in determining ITCD.

The MHFP without modification significantly varied from CIW and ITCD in both sexes, which suggested that the 3 facial proportions were unreliable in predicting MAT width. However, modified IPD values were significantly different from ITCD, except in IPD-GP female cohort where no significant differences were observed. There was also a significant difference when IPD values were compared with CIW in this study which endorses it as unreliable in MAT width determination. Furthermore, bizygomatic distance, both with and without modification, appeared to be significantly different from the MAT rendering BZD a poor predictor.

An assessment of study findings after the application of four modification metrics rendered 70% RED as an unreliable and inaccurate method for determining MAT width. Of all the studied modification metrics, recurrent esthetic dental proportion exhibited the greatest variability except with respect to IPD where IPD by 70% RED had a smaller variation than other proportions. Consequently, the mean anterior tooth dimensions predicted using 70% RED are considered unreliable. Similar findings have been reported in the literature amongst different populations, suggesting that the RED proportion does not occur in natural dentition [23,24,25,26].

The values projected by the GP were also inconsistent and did not compare well with the MAT dimensions except in “IPD by GP female group”. In contrast to other MHFP in both males and females. The BZD by GP values were constant in both sexes but significantly greater than MAT dimensions. This finding is advocated by studies evaluating the GP in a natural smile, suggesting that this proportion was not found in the successive widths of natural dentition [27,28,29].

In the present study, for most of the modification group, the Preston proportion showed disparities with both CIW and ITCD. To the best of our knowledge, the Preston proportion has not been investigated with midfacial proportions. A lack of data, therefore, makes the comparison of our outcomes rather difficult. Similarly, when assessed in natural dentition directly, PRP values of 66% and 84% between natural teeth were not seen [30,31].

In the present study, golden percentage appeared to be the only metric that showed no significant difference in predicting CIW width in both males and females. This finding is comparable to other studies. Evaluating GM in natural teeth and suggesting that if it is adjusted according to race and ethnicity, GM can serve as a reliable predictor for anterior teeth dimensions [32,33]. However, in the present study, in an attempt to predict central incisor width or inter-canine distance, modifying IPD and BZD by GM values varied significantly from the mean MAT width. This suggests that GM may not serve as a reliable predictor for maxillary anterior tooth width determination while using BZD and IPD.

Despite the strengths and uniqueness, this study also has limitations. The study subjects were categorized on the basis of race only, and not ethnicity. Tooth height was not considered. It could help in crown width-to-height ratio determination, which would have helped in categorizing the anterior teeth into small, medium, and large sizes. Only Pakistani citizens from three generations were included as study subjects. If Pakistani residents had been considered, the sample size would have been larger. A range of dental proportion values was not considered. Instead, MAT width was predicted using fixed values, i.e., golden proportion 62%, 70% Recurring esthetic dental proportion, golden percentage 10%, 15%, and 25%, and Preston proportion 66% and 84%. If a range of values had been used, it would have perhaps resulted in a reliable estimation of MAT width.

Aside from these limitations, this study revolutionizes the tooth selection process by introducing valid metrics to allow the use of dental proportions in a novel way. This is a radical concept in facial anthropometry. Although the GM and GP by ICD and IPD modification provide reliable results of CIW and ITCD, the complexity of mathematical calculations might hinder the application of this study’s methods. Further research is, therefore, required to develop a convenient and simple method using the concept and outcome of this study.

## 5. Conclusions

This study describes that:The IPD, ICD, and BZD could not directly help in determining the central incisor’s width and intercanine distance.The inner-canthal distance modified by the golden percentage can be used to determine CIW.The inter-pupillary distance modified by the golden proportion can reliably determine CIW in females.The Preston proportion and 70% RED proportion did not serve as reliable predictors of CIW and ITCD.A significant difference was found in both sexes when IPD, ICD, and BZD values were modified with GP, PRP, and 70% RED proportion. However, the ICD by golden percentage values were found to be similar in both sexes.The IPD was significantly correlated with gender in this study, while the BZD values varied significantly with the weight of participants.

## Figures and Tables

**Figure 1 jcm-11-07340-f001:**
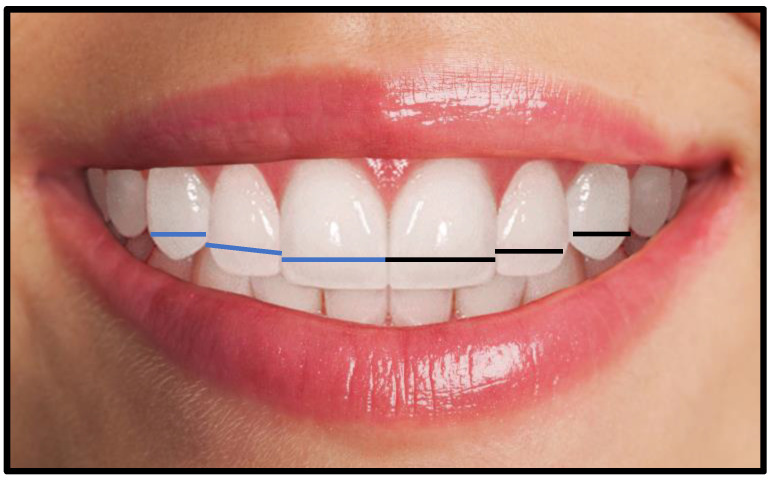
The measurements of 2-D photographic mesiodistal maxillary anterior teeth width. The blue line (right maxillary quadrant), black line (left maxillary quadrant).

**Figure 2 jcm-11-07340-f002:**
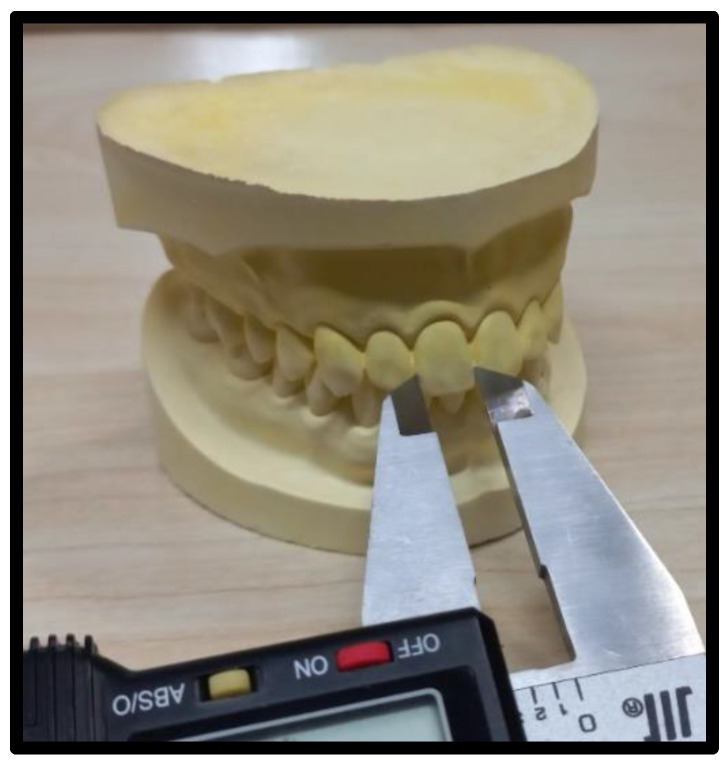
The dental stone cast mesiodistal teeth width measurements with a Vernier caliper.

**Figure 3 jcm-11-07340-f003:**
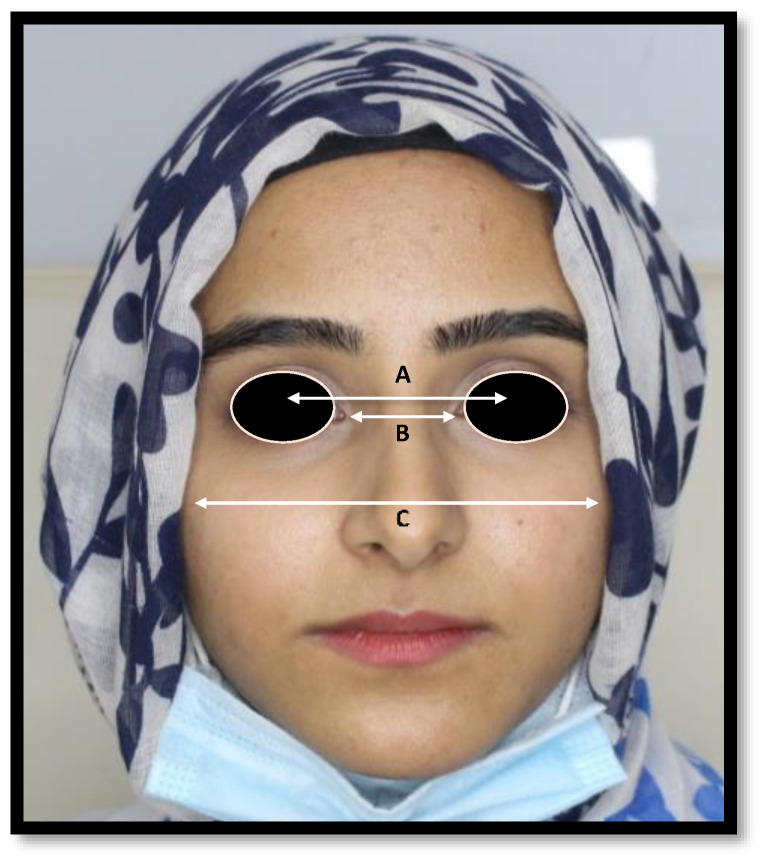
The mid horizontal facial third measurements (A) IPD (Interpupillary distance) (B) ICD (Inner canthal distance), (C) BZD (Bizygomatic distance).

**Table 1 jcm-11-07340-t001:** The distribution of modification metrics used to determine the inter-canine distance and central incisors’ width.

Dental Proportions (DP)	Horizontal Facial Third Proportion Modification Metrics
70% Recurring esthetic dental proportion (RED)	IPD or ICD or BZD × 0.70
Golden proportion (GP)	IPD or ICD or BZD × 0.62 or 1.618
Preston proportion (PRP)	IPD or ICD or BZD × 0.66^2^ (1.32) and 0.84^2^ (1.68).
Golden percentage (GM)	IPD or ICD or BZD × 0.5 or 0.3 or 0.2

**Table 2 jcm-11-07340-t002:** The distribution of mean maxillary anterior teeth widths obtained from 2D photographs, 3D models, and dental stone cast (*n* = 230).

Maxillary Teeth	2D Photographic Width		3D Digital Model Width		Dental Stone Cast Width	
	Mean (mm)	Standard Deviation	Mean (mm)	Standard Deviation	Mean (mm)	Standard Deviation
Right central incisor	16.114	2.366	8.397	0.540	8.627	0.453
Right lateral incisor	13.888	5.156	7.735	0.554	7.371	0.539
Right Canine	11.079	3.093	8.042	0.390	7.864	0.457
Left Central incisor	16.366	5.655	8.788	0.426	8.723	0.479
Left lateral incisor	13.308	1.318	7.847	0.620	7.623	0.637
Left canine	10.937	0.803	8.157	0.464	7.959	0.482
Combine six teeth width (ITCD)	81.722	9.924	48.969	1.508	48.170	1.551

2D: Two-dimensional, 3D: Three-dimensional, *p* ≤ 0.05 considered as significant through paired *t*-test analysis, ITCD: intercanine distance.

**Table 3 jcm-11-07340-t003:** The comparison of mean maxillary anterior teeth widths obtained from 2D photographs and clean width obtained after photographic error assessment (*n* = 230).

Maxillary Anterior Teeth	2D Photographic Width		Clean Width		*p*-Value
	Mean (mm)	Standard Deviation	Mean (mm)	Standard Deviation	
Right central incisor	16.114	2.366	8.130	0.717	0.001
Right lateral incisor	13.888	5.156	6.241	0.903	0.001
Right Canine	11.079	3.093	6.619	1.319	0.001
Left central incisor	16.366	5.655	7.965	0.848	0.001
Left lateral incisor	13.308	1.318	5.983	0.937	0.014
Left canine	10.937	0.803	6.384	1.320	0.027
Intercanine distance	81.722	9.924	40.788	4.090	0.001

Clean width: mesiodistal teeth dimension obtained after photographic error estimation assessment, 2D: Two dimensional, (*p* ≤ 0.05) was considered as statistically significant.

**Table 4 jcm-11-07340-t004:** The comparison between mesiodistal width of maxillary anterior teeth obtained through 3D models, in both sexes (independent *t*-test analysis (Male *n* = 112 Female *n* = 118).

Maxillary Anterior Teeth	Gender	Mean (mm)	St. Deviation	*p*-Value	*t*-Value	Mean Difference	Std. Error Difference
Right central incisor	Male	8.342	0.616	0.138	−1.490	−0.105	0.071
	Female	8.448	0.454				
Right lateral Incisor	Male	7.650	0.526	0.022	−2.305	−0.166	0.072
	Female	7.816	0.569				
Right Canine	Male	8.060	0.408	0.502	0.672	0.034	0.051
	Female	8.025	0.373				
Left central incisor	Male	8.801	0.433	0.651	0.453	0.025	0.056
	Female	8.776	0.421				
Left lateral incisor	Male	7.863	0.554	0.700	0.386	0.031	0.081
	Female	7.831	0.678				
Left canine	Male	8.186	0.477	0.361	0.915	0.056	0.061
	Female	8.130	0.452				
Intercanine distance	Male	48.905	1.511	0.531	−0.627	−0.125	0.199
	Female	49.030	1.510				

Level of significance was set at *p ≤* 0.05 and confidence interval 95%, *n* denotes: number of participants, 3D: Three dimensional, std deviation: Standard deviation: mm: millimeter, t-value: it measures the size of the difference relative to the variation in sample data, the smaller the t-value, the more similarity exists between the two sample sets. While a large t-score indicates that the groups are different. Mean Difference: the difference between the mean values from two data groups. Std. Error Difference: The standard error of the mean, it measures the variability of the sample mean, the smaller the standard error of the mean, the more likely that our sample mean is close to the true participants’ mean.

**Table 5 jcm-11-07340-t005:** The comparison of gender disparity in clean mesiodistal width of maxillary anterior teeth, independent *t*-test analysis (Male *n* = 112 Female *n* = 118).

Maxillary Anterior Teeth	Gender	Mean (mm)	St. Deviation	*p*-Value	*t*-Value	Mean Difference	Std. Error Difference
Right central incisor	Male	8.075	0.743	0.261	−1.125	−0.106	0.094
	Female	8.182	0.691				
Right lateral Incisor	Male	6.117	0.913	0.043	−2.033	−0.240	0.118
	Female	6.358	0.880				
Right Canine	Male	6.367	1.404	0.004	−2.872	−0.492	0.171
	Female	6.859	1.190				
Left central incisor	Male	7.854	0.931	0.053	−1.945	−0.216	0.111
	Female	8.071	0.748				
Left lateral incisor	Male	5.885	0.960	0.124	−1.545	−0.190	0.123
	Female	6.076	0.909				
Left canine	Male	6.072	1.337	0.001	−3.582	−0.608	0.169
	Female	6.680	1.239				
Intercanine distance	Male	39.912	4.057	0.001	−3.228	−1.706	0.528
	Female	41.619	3.961				

The level of significance was set at *p* ≤ 0.05 and confidence interval 95%, *n* denotes: the number of participants, std deviation: Standard deviation value, clean width: mesiodistal teeth dimension obtained after photographic error estimation assessment, *t*-value: It measures the size of the difference relative to the variation in sample data, the smaller the *t*-value, the more similarity exists between the two sample sets. While a large t-score indicates that the groups are different. Mean Difference: the difference between the mean values from two data groups. Std. Error Difference: The standard error of the mean, measures the variability of the sample mean, the smaller the standard error of the mean, the more likely that our sample mean is close to the true participants’ mean.

**Table 6 jcm-11-07340-t006:** The distribution of mean horizontal facial proportions without modification (*n* = 230).

Facial Proportions	Mean (mm)	Standard Deviation
Interpupillary distance (IPD)	69.233	13.319
Inner intercanthal distance	33.947	4.470
Bi-zygomatic distance	103.512	9.673
Intercanine distance *	45.976	1.784
Central incisor width *	16.909	0.697

* The intercanine distance and combined central incisor width were obtained from mean values of clean, plaster, and 3D dental cast teeth widths calculated in the study. A *p*-value ≤ 0.05 was considered significant.

**Table 7 jcm-11-07340-t007:** The comparison of mid-horizontal facial proportion in both sexes, independent *t*-test analysis (Male *n* = 112 Female *n* = 118).

Maxillary Anterior Teeth	Gender	Mean	St. Deviation	*p*-Value	*t*-Value	Mean Difference	Std. Error Difference
Interpupillary distance	Male	66.615	15.532	0.003	−2.953	−5.102	1.728
	Female	71.717	10.273				
Inner-intercanthal distance	Male	33.631	4.393	0.296	−1.047	−0.617	0.589
	Female	34.248	4.540				
Bi-zygomatic distance	Male	103.313	11.768	0.762	−0.303	−0.387	1.278
	Female	103.700	7.188				

Level of significance was set at *p* ≤ 0.05 and the confidence interval was 95%, n denotes: number of participants, std deviation: Standard deviation value, *t*-value: it measures the size of the difference relative to the variation in sample data, the smaller the *t*-value, the more similarity exists between the two sample sets. While a large t-score indicates that the groups are different. Mean Difference: the difference between the mean values from two data groups. Std. Error Difference: The standard error of the mean, it measures the variability of the sample mean, the smaller the standard error of the mean, the more likely that our sample mean is close to the true participants’ mean.

**Table 8 jcm-11-07340-t008:** The comparison of modified central incisors width and intercanine distance using dental proportions by horizontal facial proportions with measured width of central incisors and intercanine distance-mixed group (*n* = 230).

Variables	70% Recurring Aesthetic Dental Proportion	Golden Proportion	Preston Proportion		Golden Percentage	
	Modified Values					
	Mean and SD	Mean and SD	Mean and SD		Mean and SD	
			CIW	ITCD	CIW	ITCD
Interpupillary distance	48.463 ± 9.323	42.924 ± 8.257	45.693 ± 8.790	74.771 ± 4.384	34.616 ± 6.659	60.001 ± 11.543
Inner-intercanthal distance	23.763 ± 3.129	21.047 ± 2.771	22.405 ± 2.950	36.663 ±4.828	16.973 ± 2.235 ^a^	29.421 ± 3.874
Bi-zygomatic distance	72.458 ± 6.771	64.177 ± 5.997	68.317 ± 6.384	111.793 ± 10.447	51.756 ± 4.836	89.710 ± 8.383
**Measured Values**						
Central incisor width	16.909 ± 0.697 ^a^					
Intercanine distance	45.976 ± 1.784					

SD: standard deviation, CIW: combined central incisors width, ITCD: intercanine distance. ^a^ Similar superscript alphabet denotes ICD modified by GM matches with CIW.

**Table 9 jcm-11-07340-t009:** The comparison of modified central incisor width and intercanine distance using dental proportion by horizontal facial proportions with measured central incisor width and intercanine distance -male group (*n* = 112).

Variables	70% Recurring Aesthetic Dental Proportion	Golden Proportion	Preston Proportion		Golden Percentage	
	Modified Values					
	Mean (mm) and SD	Mean (mm) and SD	Mean (mm) and SD		Mean (mm) and SD	
			CIW	ITCD	CIW	ITCD
Interpupillary distance	46.630 ± 10.872	41.301 ± 9.630	43.966 ± 10.251	71.94 ± 16.775	33.307 ± 7.766	57.733 ± 13.461
Inner-intercanthal distance	23.541 ± 3.075	20.851 ± 2.723	22.196 ± 2.899	36.321 ± 4.744	16.815 ± 2.196 ^f^	29.147 ± 3.807
Bi-zygomatic distance	72.319 ± 8.237	64.054 ± 7.296	68.186 ± 7.767	111.578 ± 12.710	51.656 ± 5.884	89.538 ± 10.199
**Measured Values**						
Central incisors width	16.909 ± 0.697 ^f^					
Intercanine distance	45.976 ± 1.784					

SD: standard deviation, CIW: combined central incisors width, ITCD: intercanine distance. ^f^ Similar superscript alphabet denotes ICDGM matches with CIW, *p*-value < 0.05 was considered significant.

**Table 10 jcm-11-07340-t010:** The comparison of modified central incisor width and intercanine distance using dental proportions by horizontal facial proportions with measured central incisor width and intercanine distance -female group (*n* = 118).

Mid Horizontal Facial Proportion	70% Recurring Aesthetic Dental Proportion	Golden Proportion	Preston Proportion		Golden Percentage	
	Modified Values					
	Mean (mm) and SD	Mean (mm) and SD	Mean (mm) and SD		Mean (mm) and SD	
			CIW	ITCD	CIW	ITCD
Interpupillary distance	50.20 ± 7.191	45.465 ± 6.369 ^β^	47.333 ± 6.780	77.455 ± 11.095	35.858 ± 5.136	62.155 ± 8.903
Inner intercanthal distance	23.97 ± 3.178	21.233 ± 2.815	22.603 ± 2.996	36.988 ± 4.903	16.124 ± 2.270 ^b^	29.681 ± 3.935
Bi-zygomatic distance	72.59 ± 5.031	64.294 ± 4.456	68.442 ± 4.744	111.996 ± 7.763	51.850 ± 3.594	89.874 ± 6.229
**Measured Values**						
Combined central incisor width	16.909 ± 0.697 ^b^					
Intercanine distance	45.976 ± 1.784 ^β^					

SD: standard deviation, CIW: combined central incisors width, ITCD: intercanine distance. ^b^ Similar superscript small alphabets denote matched predicted IPD by golden proportion value with measured intercanine distance. ^β^ Similar superscript symbol denotes matched ICD by golden percentage value with measured central incisors width.

**Table 11 jcm-11-07340-t011:** The linear regression analysis of age, gender, height, and weight with mid horizontal facial third proportion (*n* = 230).

Dependent Variables	Independent Variables	Unstandardized Coefficients		Standardized Coefficients Beta (B)	*t*	*p*-Value	95% Confidence Interval for B		Collinearity	
		B_o_	Std. Error				Lower Bound	Upper Bound	Tolerance	VIF
IPD	Age	−0.022	0.246	−0.006	−0.090	0.928	−0.508	0.463	0.993	1.007
	Gender	4.834	1.943	0.182	2.487	0.014 *	1.004	8.663	0.798	1.253
	Height	0.045	0.060	0.050	0.749	0.454	−0.073	0.162	0.963	1.038
	Weight	−0.045	0.074	−0.044	−0.609	0.543	−0.190	0.100	0.821	1.218
ICD	Age	−0.025	0.084	−0.020	−0.296	0.767	−0.190	0.140	0.993	1.007
	Gender	0.410	0.661	0.046	0.619	0.536	−0.894	1.713	0.798	1.253
	Height	−0.027	0.020	−0.090	−1.340	0.182	−0.067	0.013	0.963	1.038
	Weight	−0.006	0.025	−0.017	−0.239	0.811	−0.055	0.043	0.821	1.218
BZD	Age	0.141	0.181	0.051	0.778	0.437	−0.215	0.496	0.993	1.007
	Gender	−0.973	1.424	−0.050	−0.683	0.495	−3.780	1.834	0.798	1.253
	Height	−0.025	0.044	−0.038	−0.564	0.573	−0.111	0.062	0.963	1.038
	Weight	−0.114	0.054	−0.154	−2.119	0.035 *	−0.220	−0.008	0.821	1.218

IPD = Interpupillary distance, ICD = Inner intercanthal distance, BZD = Bizygomatic distance, B = denotes the correlation between dependent and independent variables, B_0_= unstandardized coefficient, i.e., average estimation of age, gender, height, and weight with IPD, ICD, and BZD, VIF = Variance inflation factor, denotes the amount of multicollinearity in model, *t* = test of the regression coefficients, * *p*-value < 0.05, For IPD the constant for R-Squared (R^2^) = 0.041 and Adjusted R-Squared (AR^2^) was 0.0241 while for ICD (R^2^) = 0.014 and AR^2^ was −0.004. While for BZD the constant for R-Squared (R^2^) = 0.023 and Adjusted R-Squared (AR^2^) was 0.006.

## Data Availability

The data included in the present study are available upon request from the corresponding author.

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
