# Peer review of "The Analysis of Facio-Dental Proportions to Determine the Width of Maxillary Anterior Teeth: A Clinical Study"

_jcm, 2022, doi:10.3390/jcm11247340_

Round 1
Reviewer 1 Report
Comments to the Author
1. The paper is written in a good way and almost perfect for the Grammar and readability.
2. The study results are significant enough to be worth reading about and have a high impact on Dentistry field
3. Background
Well-written, indicate a gap, raise a research question, or challenge prior work in this territory. The author(s) carefully explain the goal of the study and announce the current research, clearly indicating what is original and why it is relevant, and they include up-to-date references.
Minor revision: Author did not mentioned about the gender and race variables which can bias the result, as this research use Pakistan subject, is there any differences of IPD, ICD, and BZD, MAT, GP, PRP, GM, and RED proportion from one to other Races?
4. Methods
· Fairly well written with minor typographical errors.
· Author(s) provided sufficient detail in their explanation of how the results were obtained so that an independent researcher might recreate the results adequately to allow validation of their results.
· Concerns about ethics have already been raised.
· Figure 1. Measurements of 2-D photographic mesiodistal maxillary anterior teeth width.--> could you provide a higher quality of this Figure?
· Vernier caliper positioned between the contact points of each toothà The accuracy level of the Vernier Caliper did not mention
5. Results, Discussion and Conclusions
- The results presented clearly, accurately, and presented match the methods
- Table footnote: 2D: Two-dimensional, 3D: Three dimensional, (p≤0.05) considered as significant, ITCD: intercanine distance. It should be 2D: Two-dimensional, 3D: Three dimensional, p≤0.05 considered as significant, ITCD: intercanine distance and add the information about the statistical analysis used in the footnote as well
- The Author(s) logically explained the findings and compared the findings with current findings in the research field
- The implications of the findings for future research and potential applications were clearly mentioned
- Conclusion:
4. A significant difference was found in both sexes when IPD, ICD, and BZD values were 465
modified with DP, except in the case of ICD with golden percentage. 466
4. The Preston proportion and 70% RED proportion did not serve as reliable predictors of 467
CIW and ITCD.
Is it typo? 4 after 4
Sincerely,
********
Author Response
Point-to-point author team response to reviewer comments
Dear Editor,
We are referring to your email dated 16/11/2022, for the revision of the manuscript. Thank you for reviewing our article. In response to the comments by your respective reviewer, we are sending point-to-point corrections, as per the reviewer’s comments. The correction and author’s responses that have been made are highlighted in Yellow in the attached manuscript file. All the comments have been reviewed and approved by all authors.
Author’s Response to Reviewer 1 Comments:
- The paper is written in a good way and almost perfect for the Grammar and readability.
Authors response: Thank you, we appreciate the encouragement.
- The study results are significant enough to be worth reading about and have a high impact on Dentistry field
Authors response: Thank you, we are humbled.
- Background:
Well-written, indicate a gap, raise a research question, or challenge prior work in this territory. The author(s) carefully explain the goal of the study and announce the current research, clearly indicating what is original and why it is relevant, and they include up-to-date references.
Authors response: Thank you, the author’s team is grateful.
Minor revision: Author did not mention about the gender and race variables which can bias the result, as this research use Pakistan subject, is there any differences of IPD, ICD, and BZD, MAT, GP, PRP, GM, and RED proportion from one to other Races?
Authors response: Thank you corrected, introduction section, 1st paragraph, line number 49-54, main document.
- Methods:
Fairly well written with minor typographical errors.
Authors response: Thank you, the correction is done in methods section, main document.
- Author(s) provided sufficient detail in their explanation of how the results were obtained so that an independent researcher might recreate the results adequately to allow validation of their results.
Authors response: Thank you, for kind comments, the feedback is encouraging.
- Concerns about ethics have already been raised.
Authors response: Thank you, for the comments.
- Figure 1. Measurements of 2-D photographic mesiodistal maxillary anterior teeth width.--> could you provide a higher quality of this Figure?
Authors response: Thank you, figure 1, has been replaced.
- Vernier caliper positioned between the contact points of each tooth. The accuracy level of the Vernier Caliper did not mention.
Authors response: Thank you, the information on vernier caliper accuracy and calibration is added to the text in methods section, page number 4, lines number 167 -170.
- Results, Discussion and Conclusions
The results presented clearly, accurately, and presented match the methods
Table footnote: 2D: Two-dimensional, 3D: Three dimensional, (p≤0.05) considered as significant, ITCD: intercanine distance. It should be 2D: Two-dimensional, 3D: Three dimensional, p≤0.05 considered as significant, ITCD: intercanine distance and add the information about the statistical analysis used in the footnote as well
Authors response:
The Author(s) logically explained the findings and compared the findings with current findings in the research field
Authors response: Thank you for the comment and appreciation.
The implications of the findings for future research and potential applications were clearly mentioned
Authors’ response: The authors appreciated the feedback.
Conclusion:
- A significant difference was found in both sexes when IPD, ICD, and BZD values were 465 modified with DP, except in the case of ICD with a golden percentage. 466
Authors response: The IPD , ICD, and BZD values varies in both sexes after modification with Golden proportion, golden percentage, 70% RED proportion, and Prestion proportion.
The only exception was when ICD modified with golden percentage values were compared in both sexes, we found no difference. This Means ICD values showed differences in other proportions except GM.
- The Preston proportion and 70% RED proportion did not serve as reliable predictors of 467 CIW and ITCD. Is it typo? 4 after 4
Authors response: Thank you, both proportions failed to determine the maxillary anterior teeth width i.e., CIW and ITCD.
The paper is revised, and comments are addressed, we are hopeful the revisions are sufficient to publish our paper. Thank you

Reviewer 2 Report
Thank you for submitting the manuscript.
This research work was based on one race to determine maxillary anterior width using facio-dental proportions. A convenient sample size was collected and analyzed with 2D photographs, 3D casts, and 3D images.
Some comments:
- P1, line 34. Please define the term RED in abstract section.
- P2, line 64. Please define ICD.
- P2, line 83. Please rephrase this statement and confirm the proportion of comparison.
- P2, line 87. Please define RED.
- P2, line 93. Please define MHFP.
- P3, line 115. Please define performa, and clarify its usage.
- P3, Line 142. Please clarify the validity of using 2D photographic approach to measure the canine width, and the inter-canine distance, as the canine is typically position in the curve portion of the arch.
- P4, line 145. Please provide a higher resolution of the photo for figure 1.
- P4, line 159. Please provide a punctuation after the sentence.
- P4, line 164, line 167. Please consider use the correct terminology. Is it a plaster, or stone cast? Was the plaster used to pour the impression, or only the type 4 dental stone?
- P4, line 160. Did the authors consider using scanner to scan the dentition intra-orally to obtain a 3D images directly?
- P4, line 174. Please define how BZD was measured.
- P5, line 179. Was the mandibular arch information also captured? If so, please include it in the materials and methods section.
- P6, line 233. Please insert the “maxillary” in the sentence.
- P7, line 261. Please define RLI. Line 263, please define LCI. Line 264, please define LCa and LLI.
- Did the authors consider using BMI, instead of the weight and height variables?
- P13, line 391. Please consider modifying the statement. The current study has almost equal number of female (n=118) and male (n=112) participants.
Author Response
Point-to-point author team response to reviewer comments
Dear Editor,
We are referring to your email dated 16/11/2022, for the revision of the manuscript. Thank you for reviewing our article. In response to the comments by your respective reviewer, we are sending point-to-point corrections, as per the reviewer’s comments. The correction and author’s responses that have been made are highlighted in Yellow in the attached manuscript file. All the comments have been reviewed and approved by all authors.
Reviewer 2 Review Report:
This research work was based on one race to determine maxillary anterior width using facio-dental proportions. Convenient sample size was collected and analyzed with 2D photographs, 3D casts, and 3D images.
Authors response: Thank you, we are humbled and delighted to see the reviewer’s response. Indeed, this paper has novelty and a lot of hard work from authors’ team.
Some comments:
P1, line 34. Please define the term RED in the abstract section.
Authors response: Thank you, corrected.
P2, line 64. Please define ICD.
Authors response: Thank you, corrected
P2, line 83. Please rephrase this statement and confirm the proportion of comparison.
Authors response: Thank you, the correction is done in the introduction section, lines number 85-88, and the following amendment.
“The PRP was evaluated in the width of MAT through direct and perceived measurement of teeth utilizing dental cast and 2D photographs. It was rarely found in the width of MAT [17,18]. However, as far as the determination of CIW and ITCD through modification of mid-horizontal facial proportion (MHFP) is concerned, it is yet to be studied”.
P2, line 87. Please define RED.
Authors response: Thank you, corrected.
P2, line 93. Please define MHFP.
Authors response: Thank you, corrected
P3, line 115. Please define Performa, and clarify its usage.
Authors response: Thank you, corrected.
P3, Line 142. Please clarify the validity of using 2D photographic approach to measure the canine width, and the inter-canine distance, as the canine is typically position in the curve portion of the arch.
Authors response: Thank you, corrected, the methods section, line number 146 to 150
P4, line 145. Please provide a higher resolution of the photo for figure 1.
Authors response: Thank you, corrected, figure 1 replaced, in the methods section, main document.
P4, line 159. Please provide punctuation after the sentence.
Authors response: Thank you, corrected.
P4, line 164, line 167. Please consider use the correct terminology. Is it a plaster, or stone cast? Was the plaster used to pour the impression, or only the type 4 dental stone?
Authors response: Thank you, corrected it was stone cast, plaster was used to construct the base.
P4, line 160. Did the authors consider using scanner to scan the dentition intra-orally to obtain a 3D images directly?
Authors’ response: Thank you, indirect scanning method was used in this study to obtain 3D models. To obtain the 3D models the dental stone cast after fabrication was scanned on a desktop scanner (UPCAD, UP3D).
P4, line 174. Please define how BZD was measured.
Authors response: Thank you, corrected, lines 186 to 190, methods section, main document.
P5, line 179. Was the mandibular arch information also captured? If so, please include it in the materials and methods section.
Author’s response: Thank you, only the maxillary arch data was obtained in this study, through impressions, photographs, and 3D models. The mandibular arch was not the point of focus. Hence lower arch data and details were not included in the paper
P6, line 233. Please insert the “maxillary” in the sentence.
Authors response: Thank you, corrected.
P7, line 261. Please define RLI. Line 263, please define LCI. Line 264, please define LCa and LLI.
Authors response: Thank you, corrected, lines 283 – 286, results section, main document.
Did the authors consider using BMI, instead of the weight and height variables?
Authors response: Thank you, BMI was not considered. The weight and height of participants were included for analysis in this study.
P13, line 391. Please consider modifying the statement. The current study has an almost equal number of female (n=118) and male (n=112) participants.
Authors response: Thank you, the correction is done. The following changes are performed in the text, discussion section 1st, paragraph.
“This difference in findings of the present study could be attributed to bias in sample size calculation and overestimation of the results of reported studies. Furthermore, the inappropriate allocation of male-female ratio could be another reason for dissimilar findings, because the current study has an almost equal number of female (n=118) and male (n=112) participants”.
The paper is revised, and comments are addressed, we are hopeful the revisions are sufficient to publish our paper. Thank you

Round 2
Reviewer 2 Report
Thank you for modifying and re-submitting the manuscript.
Some additional comments:
- P2, lines 90-91. Please consider modifying the statement. Could it be the lateral incisor’s width should be 66% of the central incisor width?
- P5, line 182. Please consider modifying the subtitle statement, perhaps dropping the word “plaster”? Please consider revising the paragraph accordingly.
- P14, line 439. Please consider removing the statement of “inappropriate allocation of male to the female ratio”… This study had a similar number of male (n=112) and female (n=118) participants.
Author Response
Point-to-point author team response to reviewer comments
Dear Editor,
We are referring to your email dated 02/12/2022, for the revision of the manuscript. Thank you for reviewing our article. In response to the comments by your respective reviewer, we are sending point-to-point corrections, as per the reviewer’s comments. The correction and author’s responses that have been made are highlighted in Yellow in the attached manuscript file. All the comments have been reviewed and approved by all authors.
Second reviewer, round 2 Report:
Thank you for modifying and re-submitting the manuscript.
Some additional comments:
P2, lines 90-91. Please consider modifying the statement. Could it be the lateral incisor’s width should be 66% of the central incisor width?
Authors response: Thank you, corrected
P5, line 182. Please consider modifying the subtitle statement, perhaps dropping the word “plaster”? Please consider revising the paragraph accordingly.
Authors response: Thank you, corrected, the methods section, line number 178-184, main document.
P14, line 439. Please consider removing the statement of “inappropriate allocation of male to the female ratio”… This study had a similar number of male (n=112) and female (n=118) participants.
Authors response: Thank you, the sentence is omitted from 1st paragraph, discussion section.
The paper is revised, and comments are addressed, we are hopeful the revisions are sufficient to publish our paper. Thank you
